# Characterization of Two Novel Insect-Specific Viruses Discovered in the Green Leafhopper, *Cicadella viridis*

**DOI:** 10.3390/insects13040378

**Published:** 2022-04-12

**Authors:** Li-Li Li, Zhuang-Xin Ye, Jian-Ping Chen, Chuan-Xi Zhang, Hai-Jian Huang, Jun-Min Li

**Affiliations:** State Key Laboratory for Managing Biotic and Chemical Threats to the Quality and Safety of Agro-Products, Key Laboratory of Biotechnology in Plant Protection of Ministry of Agriculture and Zhejiang Province, Institute of Plant Virology, Ningbo University, Ningbo 315211, China; 2011074033@nbu.edu.cn (L.-L.L.); yzx244522794@163.com (Z.-X.Y.); jpchen2001@126.com (J.-P.C.); chxzhang@zju.edu.cn (C.-X.Z.)

**Keywords:** *Cicadella viridis*, insect-specific virus, metatranscriptomic sequencing, *Iflavirus*, nidovirus, antiviral RNAi

## Abstract

**Simple Summary:**

Insect-specific viruses (ISVs) have gained increasing attention for their potential use as biological agents. In this study, the full genomes of two ISVs (*Cicadella viridis iflavirus 1*, CvIfV1; *Cicadella viridis nido-like virus 1*, CvNiLV1) were revealed in green leafhoppers, using metatranscriptome, RT-PCR, and RACE approaches, respectively. CvIfV1 is a member of *iflavirus* and has a typical *iflavirus* genome organization. An antiviral RNA interference (RNAi) was triggered when host insects were challenged with CvIfV1, which resulted in an abundant accumulation of 21-nt virus-derived siRNAs (vsiRNAs). CvNiLV1 clusters are within a distinct unclassified clade of viruses in the order *Nidovirales*, and evolutionary related with viruses that infect vertebrate hosts. CvNiLV1 was also targeted by the host antiviral RNAi pathway, and generated the 21-nt vsiRNAs with a strong A/U bias in the 5′-terminal. Our study provided valuable information on ISVs in leafhoppers, and might prove useful in pest management.

**Abstract:**

Insect-specific viruses (ISV) are one of the most promising agents for the biological control of insects. The green leafhopper, *Cicadella viridis* (Linnaeus), is an significant pest in agriculture, and causes economic losses to fruit trees, willows, and field crops. As a representative species of the large family Cicadellidae, ISVs in *C. viridis* have never been studied, to date. In this study, the full genome sequences of two ISVs, named *Cicadella viridis iflavirus*
*1* (CvIfV1), and *Cicadella viridis nido-like virus 1* (CvNiLV1), were revealed using a metatranscriptomic approach. A homology search and phylogenetic analysis indicated that CvIfV1 is a new member in the family *Iflaviridae* (genus *Iflavirus*) with a typical *iflavirus* genome organization, whereas CvNiLV1 belongs to the unclassified clade/family of the order *Nidovirales*. In addition, analysis of virus-derived small interfering RNAs (vsiRNAs) was performed to investigate the antiviral RNA interference (RNAi) response of *C. viridis*. The vsiRNAs exhibit typical patterns produced by host siRNA-mediated antiviral immunity, including a preference of 21-nt vsiRNAs derived equally from the sense and antisense genomic strands, and a strong A/U bias in the 5′-terminus of the viral genomes. Our study provides valuable information for ISVs in leafhoppers for the first time, which might prove useful in the control of *C. viridis* in future.

## 1. Introduction

Insects are the largest group of organisms on earth, and insect virus resources are an essential part of natural resources. Previously, most insect viruses were discovered due to their pathogenic effects on their hosts or because they caused diseases [1,2]. However, the emergence in recent decades of metagenomic/metatranscriptomic next-generation sequencing and bioinformatics analysis has made it possible to conduct de novo sequencing and assemble numerous novel RNA viruses from non-pathogenic insects [3,4,5,6,7]. These viruses are generally referred to as insect-specific viruses (ISVs), which exclusively replicate in their insect hosts, and transovarially or vertically transmit within a population [8]. ISVs are considered as promising agents for the biological control of specific insects and the pathogens they transmit, gaining them increasing attention [9,10]. Comprehensive analysis of ISVs has been carried out on mosquitos, ticks, and other medically important hematophagous insects [11,12]. Phylogenetic analysis demonstrates that the majority of identified ISVs belong to the particular virus taxa, including *Ascoviridae*, *Baculoviridae*, *Bunyaviridae* (now *Bunyavirales*), *Flaviviridae*, *Parvoviridae*, *Rhabdoviridae*, *Togaviridae*, and novel groups as yet unclassified [13].

Herbivorous insects make up approximately 50% of all insect species, and damage nearly 18% of global agricultural production [14]. Considering their great diversity, our knowledge of ISVs in herbivorous insects is still insufficient. Cicadomorpha is an infraorder of Hemiptera, which is comprised of approximately 35,000 described species [15]. Several species of this infraorder are significant pests to crops, and vectors of plant pathogens. The green leafhopper, *Cicadella viridis* (Linnaeus), is a sap sucking, polyphagous insect and a representative species of the large family Cicadellidae, belonging to Cicadomorpha. It is cylindrical-shaped, and widely distributed in humid areas, causing economic losses to fruit trees, willows, and field crops [14]. Similar to most leafhoppers, *C. viridis* is capable of transmitting several diseases, including Pierce’s disease and the European stone fruit yellows phytoplasma [16,17]. The bacteriocyte endosymbionts residing in *C. viridis* have been discovered and provide fundamental resources for pest control [18]. However, thus far, ISVs in *C. viridis* have not been characterized. Even for insects in the Cicadomorpha infraorder, only a limited number of ISVs have been reported [19,20,21].

In the present study two ISVs, named *Cicadella viridis iflavirus*
*1* (CvIfV1) and *Cicadella viridis nido-like virus 1* (CvNiLV1), were firstly identified in *C. viridis* using metatranscriptomic technology. The presence of ISVs was then verified using reverse transcription PCR (RT-PCR), and their full genome sequences were confirmed by rapid amplification of cDNA ends (RACE). Furthermore, analysis of virus-derived small interfering RNA (vsiRNA) clearly indicated an active antiviral RNAi pathway in *C. viridis* in response to the two ISVs.

## 2. Materials and Methods

### 2.1. Sample Preparation and RNA Extraction

The green leafhoppers analyzed in the current work were collected in Changchun, Jilin, China, in September 2020. The sample was sent to our laboratory alive, and immediately homogenized in trizol reagent (Invitrogen, Waltham, MA, USA). Altogether, three adult leafhoppers (whole body) were merged into one sample, and used for RNA extraction. The extracted RNA sample was subsequently divided into three tubes, which were used for transcriptomic sequencing, small RNA (sRNA) sequencing, and full genome sequence determination.

### 2.2. Host Insect Identification

To identify the exact species of the green leafhopper, the mitochondrial cytochrome oxidase I (COI) sequence of the insect was cloned, and further confirmed by Sanger sequencing. The sequence was then compared to the Barcode of Life Data (BOLD) System (https://www.boldsystems.org/) accessed on 1 December 2020.

### 2.3. Transcriptomic and Small RNA (sRNA) Sequencing

The extracted RNA samples were sent to Novogene, Tianjin, China for transcriptomic and small RNA (sRNA) sequencing. Library preparations for transcriptome and sRNA were performed, as described previously [5]. In brief, for transcriptomic sequencing a paired end (150 bp) library was constructed and sequenced on the Illumina HiSeq 4000 platform (Illumina, San Diego, CA, USA). The outputted raw reads were quality trimmed of adaptor sequences, and the clean reads were de novo assembled, using the Trinity software (Version 2.8.5) with default parameters [22]. For sRNA sequencing, the sRNA library was prepared using the Illumina TruSeq small RNA sample preparation kit (Illumina, San Diego, CA, USA), and sequenced on the Illumina HiSeq 2500 platform. The outputted raw data were later quality trimmed by removing the adapters and low-quality sequences using the Cutadapt tool [23].

### 2.4. Virus Discovery and Confirmation by Reverse Transcription-PCR (RT-PCR)

The potential viruses in the sample were discovered according to our previously described methods [24]. Briefly, the assembled contigs were aligned against the NCBI viral RefSeq database, using the diamond BlastX with the cutoff E-value of 1 × 10^−20^. Thereafter, the viral homology contigs with sequence lengths of no less than 3000 bp were selected for further BLAST analysis against the entire NCBI nucleotide (NT) and the non-redundant (NR) protein databases, so as to avoid false positive matches. Finally, the identified virus-derived contigs were verified using RT-PCR, followed by Sanger sequencing using the primers listed in Appendix A.

### 2.5. Determination of Viral Genome Termini and Transcript Abundance

The rapid amplification of cDNA ends (RACE) technology was employed to obtain the full genome sequence of the viruses. Briefly, the 5′-RACE-ready and 3′-RACE-ready cDNA was synthesized using the SMARTer^®^ RACE 5′/3′ kit (Takara, Beijing, China), according to the manufacturer’s instructions. Subsequently, the prepared cDNA was used in the touchdown PCR using the 5′ or 3′ gene-specific primers (GSPs) and universal primer A mix (UPM). The PCR reaction (20 μL) was performed using 1 μL adaptor cDNA, 2 μL 10 × UTM primer mix, 0.8 μL GSPs, 10 μL 2× Phanata Buffer, 0.4 μL dNTP, and 1 μL Phanata Max Super-Fidelity DNA polymerase (Vazyme, Nanjing, China) under the following conditions: denaturation at 95 °C for 3 min, followed by 10 cycles of 95 °C for 30 s, 65 °C for 30 s, and 72 °C for 30 s, subsequently followed by 30 cycles of 95 °C for 30 s, 60 °C for 30 s, and 72 °C for 30 s. The primers used for RACE are listed in Appendix A. Later, the PCR products were cloned into the pClone007 vector (Tsingke, Beijing, China), followed by Sanger sequencing.

To determine the transcript coverage and the abundance of the viruses, the adaptor- and quality-trimmed reads from the RNA-seq data generated in this study were mapped back to the full viral genomes using Bowtie2 and Samtools [25,26]. The coverage of the aligned reads to the virus genomes was subsequently visualized using an Integrated Genomics Viewer. To investigate the prevalence of viruses in different samples, we searched for the RNA-seq data of *C. viridis* in public databases, and found one dataset in the NCBI SRA repository (accession number: SRR11729954). The RNA-seq reads of SRR11729954 were then mapped back to the virus genomes, as described above.

### 2.6. Small RNA Analysis

To identify the siRNAs derived from the two viruses, the 18–30 nt long quality trimmed sRNA reads were extracted and collapsed using the FASTX-Toolkit (http://hannonlab.cshl.edu/fastx_toolkit/) accessed on 1 May 2021. Later, the processed reads were mapped back to the assembled full viral genome sequence using Bowtie with zero mismatches, and the output virus-derived siRNAs (vsiRNAs) were further calculated with custom perl script and the Linux bash scripts [27].

### 2.7. Genome Annotation and Phylogenetic Analysis

The open reading frames (ORFs) were predicted based on the ExPASy (https://web.expasy.org/translate/) accessed on 1 May 2021. The conserved protein domains were predicated using the InterProScan (https://www.ebi.ac.uk/interpro) accessed on 1 May 2021. The amino acid sequences of predicated RdRp domains of iflaviruses, or RdRp proteins of nidoviruses, were retrieved from the NCBI. The sequences were aligned with MAFFT (version 7.450), and the gaps were further trimmed using Gblock [28]. The substitution model was evaluated using ModelTest-NG based on the default parameters [29]. Afterwards, maximum likelihood (ML) trees were constructed using RAxMLNG (version 0.9.0) with 1000 bootstrap replications [30].

## 3. Results and Discussions

### 3.1. Transcriptome Assembly and Virus Discovery for C. viridis

A total of 96,813 contigs were generated with the mean length of 1075 bp and the N50 of 2096 bp. The COI sequence of the insect (accession number: ON142388) was first compared with the Barcode of Life Data (BOLD) System, and a BLAST search confirmed that the species of the green leafhopper was *C. viridis*, with the COI sequence identical to the reported COI sequence of *C. viridis* in GenBank.

To identify potential viruses presented in the transcriptome, the assembled contigs were BLAST against the NCBI viral RefSeq database, and altogether, 674 contigs showed sequence similarity to viral proteins with a cutoff E-value of 1 × 10^−20^. These contigs were further used to BLAST search the entirety of the NCBI databases, and the species distributions of the best matches for each contigs are shown in Appendix A. As the results show, it is clear that the majority of contigs (672/674, 99.7%) showed a higher sequence similarity to insect proteins rather than viral proteins (Appendix A), indicating that they are not derived from exogenous viruses. After filtering host-originated contigs, two viral contigs were finally confirmed from the assembled library. Based on the taxonomy of the most closely related viruses, these two viral contigs are putative members of the family *Iflaviridae* (named *Cicadella viridis iflavirus 1*, CvIfV1), or related to the order *Nidovirales* (named *Cicadella viridis nido-like virus 1*, CvNiLV1), respectively.

### 3.2. Cicadella viridis iflavirus 1

The presence of CvIfV1 was verified using reverse transcription PCR (RT-PCR), and its full genome sequence was successfully obtained by RACE using a SMARTer^®^ RACE 5′/3′ Kit. The full genome sequence of CvIfV1 was 9716 nt in length (accession number: OM774425), and contained a 181-nt 5′ untranslated region (UTR), a 9297-nt open reading frame (ORF), and a 238-nt 3′ UTR (Figure 1A). According to the InterProScan prediction of the conserved domains, the long ORF of CvIfV1 contains the typical domains of iflaviruses, including two picornavirus-like capsid protein domains (RHV), a cricket paralysis virus capsid superfamily domain (CRPV), an RNA helicase domain (RNA_Hel), a protease domain (PRO), and an RNA-dependent RNA polymerase domain (RdRp) (Figure 1A).

CvIfV1 shared 51.69% of its identity in the coat protein (CP) region with its closest homolog, Turkana Iflavi-like virus 2 (accession number: UCW41656.1), thereby meeting the demarcation criterion (lower than 90% identity in the CP) for establishing a new species in the genus *Iflavirus*. The abundance and coverage of CvIfV1 were evaluated using a realignment of the RNA-seq reads to the reconstructed full genome sequence of CvIfV1. A total of 68,705 reads are perfectly mapped to the CvIfV1 genome, accounting for 0.37% of the whole RNA-seq reads (Figure 1B). The transcripts are distributed across the whole viral genome, with an elevated abundance in the 3′ terminus, indicating that CvIfV1 efficiently replicated in the host insect (Figure 1A). Currently, one RNA-seq dataset of *C. viridis* (SRR11729954) is available in the NCBI SRA repository. To investigate the prevalence of CvIfV1, the RNA-seq reads of SRR11729954 were rearranged into the CvIfV1 genome. The results show that 179,722 reads (0.49% of the whole RNA-seq reads) are mapped to the CvIfV1 genome (Appendix A), indicating that CvIfV1 might prevalently infect *C. viridis*. An RdRP domain-based ML tree shows that CvIfV1 is clustered with La Jolla virus, Lygus lineolaris virus 1, Sacbrood virus, and Halyomorpha halys virus, and is supported by a relatively high bootstrap value (Figure 2). Although there is only one genus in *Iflaviridae* family, phylogenetic analysis indicates that iflaviruses are divided into several distinct clades (Figure 3). These clades may be divided into different genera when more iflaviruses are found in the future. 

### 3.3. Cicadella viridis Nido-like Virus 1

The presence of CvNiLV1 was also confirmed by RT-PCR, and the complete 5′ and 3′ UTR successfully achieved by RACE, as described above. CvNiLV1 has a genome size of 20,193 nt (accession number: OM774426), which contained four typical nidovirus ORFs, a 76-nt 5′ UTR, and a 1046-nt 3′ UTR (Figure 1B). It was highly homologous to *Bemisia tabaci* nido-like virus and Wuhan insect virus 19, with the RdRp protein sequence identities of 33.57% and 36.09%, respectively. It is noteworthy that there is an overlap between ORF3 and ORF4 in CvNiLV1 (Figure 1B), which is the typical characteristic of nidovirus [31]. The abundance and coverage of CvNiLV1 were then evaluated using the RNA-seq data generated in this study, and retrieved from the NCBI SRA repository. A total 5858 reads (0.03% of the whole RNA-seq reads) from the transcriptomic data generated in this study are perfectly mapped to CvNiLV1 genome (Figure 1B). In contrast, no reads from SRR11729954 can be mapped to CvNiLV1 genome, indicating that not all *C. viridis* were infected with CvNiLV1.

According to phylogenetic analysis, CvNiLV1 is clustered within a distinct unclassified clade of viruses with insect hosts (Figure 3). This unclassified clade belongs to the order *Nidovirales*, and is most closely related to the family *Arterividae* of *Nidovirales* that infects vertebrate hosts only. According to the latest ICTV Online (10th) report, the order *Nidovirales* is classified into nine families, including *Coronaviridae*, *Abyssoviridae*, *Arteriviridae*, and *Mesnidovirineae*. All nidovirus members possess the single-stranded, positive sense RNA genomes, and rank among the most complex RNA viruses [32]. CvNiLV1 shows typical nidovirus structure, but cannot be assigned to one of the established nidovirus families. This insect-associated nidovirus clade likely represents a novel family within the order Nidovirales, which deserves further investigation. Currently, our knowledge on nidoviruses is mostly derived from studies of coronaviruses, especially the SARS-CoV-2 [33], the causative agent of the recent COVID-19 outbreak. Identification of CvNiLV1 in insect species will enrich our knowledge on nidoviruses, and might contribute to our understanding for the origin of coronaviruses. 

### 3.4. Activation of Antiviral RNA Interference Pathway in C. viridis Responsive to ISVs

siRNA-based RNA silencing is one of the most important antiviral immune responses in insects to eliminate invading viruses, and is usually correlated with the accumulation of vsiRNAs, since viral RNAs are cleaved in a sequence-specific manner. To explore the siRNA-based antiviral immunity of the *C. viridis* host, sRNA of *C. viridis* were sequenced, and the vsiRNAs were comprehensively analyzed. A total of 1872 siRNA reads (975 unique reads) perfectly mapped to CvIfV1 and CvNiLV1 genomes were obtained from a total of 18,647,549 sRNA reads library. The majority of CvIfV1-derived vsiRNAs are 21-nt in length, and almost equally derived from the antisense and sense strands of the viral genome (Figure 4A). Meanwhile, similar distribution patterns are observed for CvNiLV-derived vsiRNAs, with the 21-nt vsiRNAs being dominant (Figure 5A). Considering that the 21-nt vsiRNAs length peak was usually reported in dipterans [5], while 22-nt vsiRNAs peak was frequently observed for hemipterans [22,23], it will be intriguing to explore the Dicer-like protein responsible for the production of 21-nt vsiRNAs in *C. viridis* in any future study.

In addition, there is a strong A/U bias in the 5′-terminal nucleotide preference of the 21-nt long vsiRNAs (Figure 4C,D and Figure 5C,D), which is a typical signature of vsiRNAs in insect species. It is noteworthy that, although the vsiRNAs from both viruses are distributed across the entire viral genome, including the UTRs, there are notable asymmetric hotspots on both strands (Figure 4B and Figure 5B). This indicates that the host antiviral system preferred to target these regions for cleavage. The typical characteristics of the above-mentioned CvIfV1- and CvNiLV1-derived siRNAs strongly suggest an active involvement of the RNAi pathway of *C. viridis* in response to ISV infection.

## 4. Conclusions

This study revealed the full genomes of CvIfV1 and CvNiLV1 in leafhopper species using metatranscriptomic sequencing and RACE technology. CvIfV1 is a member of *iflavirus* and encoded a polyprotein to generate the structural and non-structural proteins. CvNiLV1 is an evolutionarily virus related to an unclassified viral clade in the order *Nidovirales*, and related to viruses that infect vertebrate hosts. Both of the viruses are targeted by host antiviral RNAi, which resulted in the accumulation of an abundance of 21-nt vsiRNAs. The results of this study contribute to a better understanding on ISVs in leafhopper species, providing valuable information on the evolution of iflaviruses and nidoviruses.

## Figures and Tables

**Figure 1 insects-13-00378-f001:**
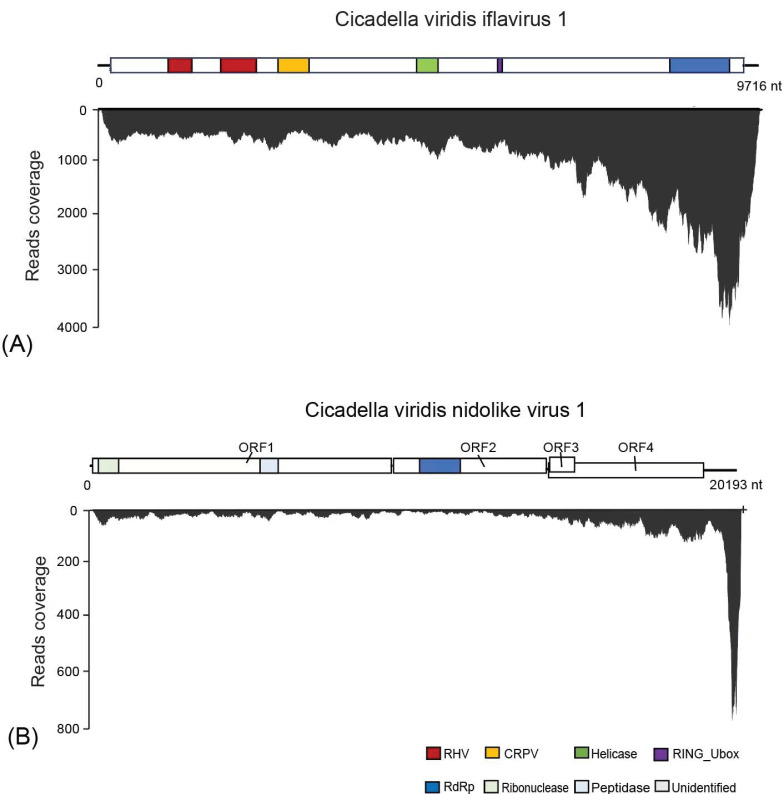
Genomic structure and RNA-seq read coverage of *Cicadella viridis iflavirus 1* (**A**) and *Cicadella viridis nido-like virus 1* (**B**). The boxes represent the open reading frames (ORFs) of each virus. Conserved functional domains are color-coded and their corresponding names are indicated at the bottom of the figure. Abbreviation of the conserved domain names: *RHV*, picornavirus capsid protein domains; *CRPV*, cricket paralysis virus capsid superfamily domain; *RdRp*, RNA-dependent RNA polymerase domain.

**Figure 2 insects-13-00378-f002:**
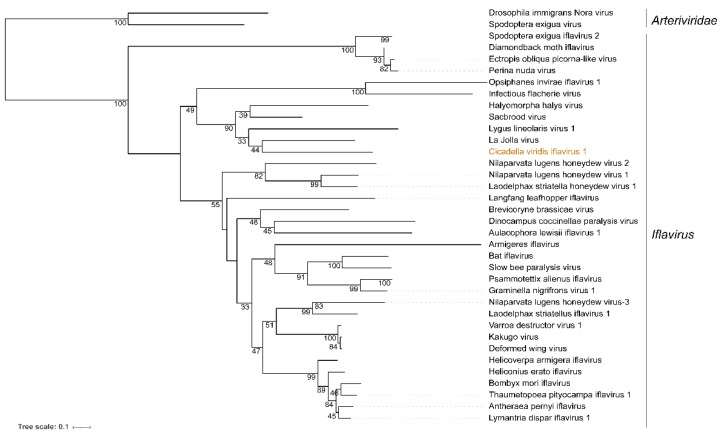
**Phylogenetic analysis of *Cicadella viridis iflavirus 1* (CvIfV1) with other iflaviruses**. Maximum likelihood phylogenetic tree based on the RNA-dependent RNA polymerase domain was constructed with a bootstrap of 1000. Scale bars represent percent divergence. Drosophila immigrans Nora virus and Spodoptera exigua virus are used as an outgroup.

**Figure 3 insects-13-00378-f003:**
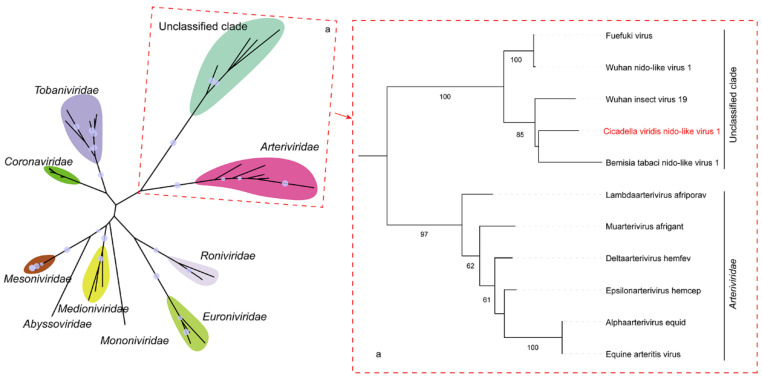
**Phylogenetic analysis of *Cicadella viridis nido-like virus 1* (CvNiLV1) with other nidoviruses**. Maximum likelihood phylogenetic tree based on the RNA-dependent RNA polymerase was constructed with a bootstrap of 1000. A taxonomic overview of viruses at order Nidovirales are shown on the left, and a close-up view of two interested clades are shown in the dotted frames on the right.

**Figure 4 insects-13-00378-f004:**
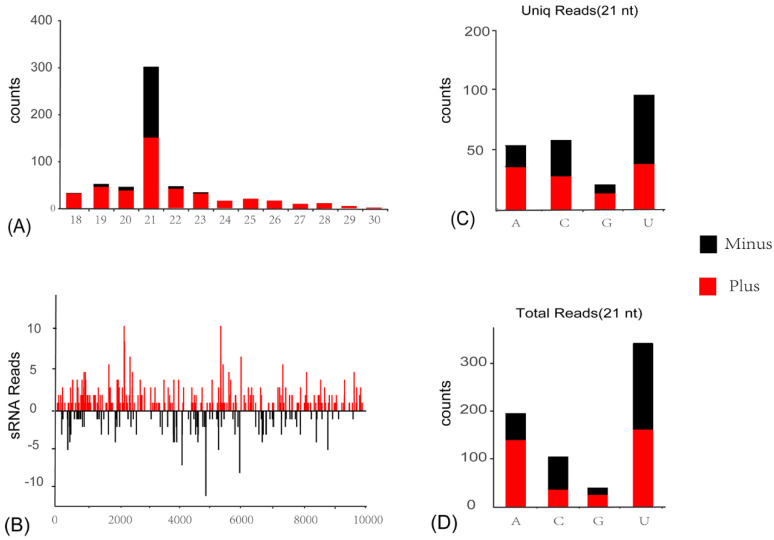
**Profiles of virus-derived small interfering RNAs (vsiRNAs) of *Cicadella viridis iflavirus 1* (CvIfV1)**. (**A**) Size distribution of CvIfV1-derived siRNA. (**B**) Distribution of CvIfV1-derived siRNA along the viral genome. (**C**,**D**) 5′-terminal nucleotide preference of 21-nt long vsiRNAs analyzed by uniq reads (**C**), and total reads (**D**).

**Figure 5 insects-13-00378-f005:**
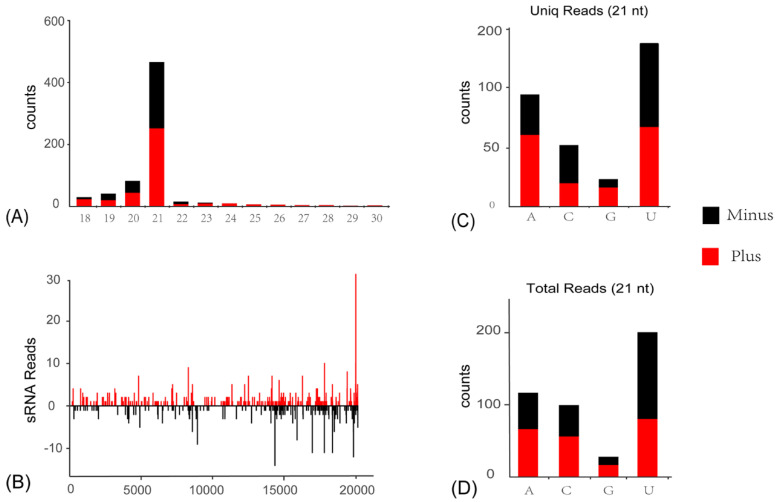
**Profiles of virus-derived small interfering RNAs (vsiRNAs) of *Cicadella viridis nido-like virus 1* (CvNiLV1)**. (**A**) Size distribution of CvNiLV1-derived siRNA. (**B**) Distribution of CvNiLV1-derived siRNA along the viral genome. (**C**,**D**) 5′-terminal nucleotide preference of 21-nt long vsiRNAs analyzed by uniq reads (**C**), and total reads (**D**).

## Data Availability

The data presented in this study are available within the article and Appendix A.

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
