# Peer review of "Characterization of Two Novel Insect-Specific Viruses Discovered in the Green Leafhopper, Cicadella viridis"

_insects, 2022, doi:10.3390/insects13040378_

Round 1
Reviewer 1 Report
The authors of the manuscript perform the meta-transciptomic analysis of leafhopper and report the findings of two viruses, CvIfV1 and CvNiLV1, which belong to unclassified clade of Nidovirus. In addition, the authors performed the vsiRNA and found an immunity system response with bias toward 21nt sviRNA production.
The manuscript is descriptive, short and straightforward to describe the findings, however, I was surprised by Discussion to be very short (in fact, merged with the Results section). It would be nice to have more context for the results, however, it could be fine for Communication.
Minor comments:
Line 9: The first sentence is not clear.
Line 54: Correct the font for the reference
Line 60: Correct the fond for “and”
Material and Methods:
Line 80: Which tissues were extracted for RNAseq? What was the sample size and sex of the analysed insects?
Line 140: Was this sentence supposed to be in Methods?
Author Response
The authors of the manuscript perform the meta-transciptomic analysis of leafhopper and report the findings of two viruses, CvIfV1 and CvNiLV1, which belong to unclassified clade of Nidovirus. In addition, the authors performed the vsiRNA and found an immunity system response with bias toward 21nt sviRNA production.
Response: Thank you very much for your comments on our manuscript.
The manuscript is descriptive, short and straightforward to describe the findings, however, I was surprised by Discussion to be very short (in fact, merged with the Results section). It would be nice to have more context for the results, however, it could be fine for Communication.
Response: Thank you for your suggestion. The manuscript was submitted as “Communication” type, and the study mainly focused on the new discoveries for the ISVs in green leafhopper. As the type of “Communication”, the discussions have been merged into the description of results.
In the revised manuscript, considering that readers might be more interested in CvNiLV1 (as this virus, together with SARS-CoV-2, belongs to the same order Nidovirales), we added discussion on the potential taxonomic status of CvNiLV1, which might represent the insect-associated nidovirus clade in the order Nidovirales. Please see Line227-237 in the revised manuscript.
Minor comments:
Line 9: The first sentence is not clear.
Response: Done. In the revised version, we rewrote this sentence as “Insect-specific viruses (ISVs) have gained increasing attention in their potential use as biological agents”. Please see Line 9-10 in the revised manuscript.
Line 54: Correct the font for the reference
Response: Done. Please see Line 55 in the revised manuscript.
Line 60: Correct the fond for “and”
Response: Done. Please see Line 60 in the revised manuscript.
Material and Methods:
Line 80: Which tissues were extracted for RNAseq? What was the sample size and sex of the analysed insects?
Response: Thank you for your suggestion. In the revised version, we addressed that “Altogether 3 adult leafhoppers (whole body) were merged into one sample, and used for RNA extraction”. As we did not distinguish the sex of the analyzed insects, we did not provide this information in the manuscript. Please see Line 79-81 in the revised manuscript.
Line 140: Was this sentence supposed to be in Methods?
Response: Thank you for your suggestion. This information has been provided in the Material and Method section (Line 77-78 in the revised manuscript). Therefore, this sentence has been deleted accordingly.

Reviewer 2 Report
Characterization of two novel insect-specific viruses discovered in the green leafhopper, Cicadella viridis
The authors have sequenced the hoppers collected from the field to identify the insect specific viruses with a rationale to identify novel pest control strategies. Transcriptome sequencing and bioinformatics tools were used to identify the virus like sequences and RACE was used to sequence the full length sequences.
The major concern with this study is the number of insects used in the study. According to the results, there are only 3 insects used for the study which can minimize the number of viruses that are identified and the probability of finding them in the samples is random. Confirming the presence of these viruses in more biological collections would support the findings. In the present form, the manuscript cannot be accepted for publication.
Line 13: Remove fierce
Line 53: This line is out of place in the entire context of the paragraph. Either move it elsewhere or delete it
Line 77: More information on the number of biological replicates and the number of insects used should be mentioned clearly
Line 85: expand sRNA
Line 88: Which package was used for quality trimming?
Line 91: Please include the details of the method here. If the approach was similar to the reference cited in here, include the details briefly.
Line 94: This section should follow the insect collection section or can be included in the same paragraph. The sequence should be deposited in NCBI and the accession number should be included in the manuscript
Line 108: Please include PCR parameters in this section
Line 122: What was the goal of the analysis with the custom perl scripts? It is not clear in this paragraph. Please include the missing details
Line 135: What were the evaluation parameters?
Author Response
The authors have sequenced the hoppers collected from the field to identify the insect specific viruses with a rationale to identify novel pest control strategies. Transcriptome sequencing and bioinformatics tools were used to identify the virus like sequences and RACE was used to sequence the full length sequences.
Response: Thank you very much for your comments on our manuscript.
The major concern with this study is the number of insects used in the study. According to the results, there are only 3 insects used for the study which can minimize the number of viruses that are identified and the probability of finding them in the samples is random. Confirming the presence of these viruses in more biological collections would support the findings. In the present form, the manuscript cannot be accepted for publication.
Response: Thank you very much for your valuable suggestion. We do agree that it is necessary and essential to confirm the presence of these viruses in more biological collections. Unfortunately, we regret that the collected leafhoppers were all used for RNA extraction, and we failed to establish the leafhopper culture in the lab at that time. Furthermore, due to the travel restrictions of SARS-COV19, we were unable to collect the leafhopper in the same location again these days.
Instead, we searched the RNA-seq dataset of Cicadella viridis in NCBI SRA repository to assess the prevalence of the discovered viruses in this study. One RNA-seq dataset of C. viridis (accession number: SRR11729954) was available in NCBI SRA repository. Then, the RNA-seq reads of SRR11729954 were mapped back to the full genomes of CvIfV1 and CvNiLV1, respectively. The results showed that 179,722 reads (0.49% of the whole RNA-seq reads) were successfully mapped to CvIfV1 genome. In contrast, no read was mapped to CvNiLV1 genome. These results indicated that CvIfV1 might prevalently infect C. viridis, while not all C. viridis were infected with CvNiLV1. Also, we provided the total reads and the percentage of reads mapped to the virus from each library, which improved our knowledge on viral abundance.
In the revised version, we describe these results as follow: “A total of 68,705 reads were perfectly mapped to CvIfV1 genome, accounting for 0.37% of the whole RNA-seq reads (Fig. 1B)”, “Currently, one RNA-seq dataset of C. viridis (SRR11729954) was available in NCBI SRA repository. To investigate the prevalence of CvIfV1, RNA-seq reads of SRR11729954 were mapped back to CvIfV1 genome. The results showed that 179,722 reads (0.49% of the whole RNA-seq reads) were successfully mapped to CvIfV1 genome, indicating that CvIfV1 might prevalently infect C. viridis”, “Then, the abundance and coverage of CvNiLV1 were evaluated using the RNA-seq data of C. viridis generated in this study and retrieved from NCBI SRA repository (SRR11729954). A total number of 5,858 reads (0.03% of the whole RNA-seq reads) from the transcriptomic data generated in this study were perfectly mapped to CvNiLV1 genome (Fig. 1B). In contrast, no reads from SRR11729954 can be mapped to CvNiLV1 genome, indicating that not all C. viridis were infected with CvNiLV1.” Please see Line 180-186, Line 230-237 in the revised manuscript.
Line 13: Remove fierce
Response: Done.
Line 53: This line is out of place in the entire context of the paragraph. Either move it elsewhere or delete it
Response: Thank you for pointing out this. In the revised version, we've moved it to the next paragraph. Please see Line 54-57 in the revised manuscript.
Line 77: More information on the number of biological replicates and the number of insects used should be mentioned clearly
Response: Done. In the revised version, we addressed that “Altogether 3 adult leafhoppers (whole body) were merged into one sample, and used for RNA extraction.” Please see Line 79-81 in the revised manuscript.
Line 85: expand sRNA
Response: Done.
Line 88: Which package was used for quality trimming?
Response: Done. In the revised version, we addressed that Cutadapt tool was used for quality trimming. Please see Line 99-100 in the revised manuscript.
Line 91: Please include the details of the method here. If the approach was similar to the reference cited in here, include the details briefly.
Response: Done. We provide the detail procedure in brief as follow: “Briefly, the assembled contigs were aligned against the NCBI viral RefSeq database using the diamond BlastX with the cutoff E-value of 1×10-20. Thereafter, the viral ho-mology contigs with sequence length no less than 3000 bp were selected for further BLAST analysis against the entire NCBI nucleotide (NT) and the non-redundant (NR) protein databases, so as to avoid false positive matches. The identified virus-derived contigs were finally verified by RT-PCR followed by Sanger sequencing using the pri-mers listed in Supplementary Table 1.”
Line 94: This section should follow the insect collection section or can be included in the same paragraph. The sequence should be deposited in NCBI and the accession number should be included in the manuscript
Response: Done. In the revised version, the section of “Host Insect Identification” has been moved following the insect collection section. Also, the COI sequence identified in this study have been deposited in NCBI under the accession number ON142388. We added this new accession number in the revised version.
Line 108: Please include PCR parameters in this section
Response: Done. In the revised version, PCR parameters for RACE are provided in the material and method section as follow: “The PCR reaction (20μL) was performed using 1 μL adaptor cDNA, 2 μL 10×UTM pri-mer Mix, 0.8 μL GSPs, 10 μL 2× Phanata Buffer, 0.4 μL dNTP, and 1 μL Phanata Max Su-per-Fidelity DNA Polymerase under the following conditions: denaturation at 95 °C for 3 min, followed by 10 cycles of 95 °C for 30 s, 65 °C for 30s, 72 °C for 30s, subsequently 30 cycles of 95 °C for 30 s, 60 °C for 30s and 72 °C for 30s.”
Line 122: What was the goal of the analysis with the custom perl scripts? It is not clear in this paragraph. Please include the missing details
Response: Done. In the revised version, we addressed that “Later, the processed reads were mapped back to the assembled full viral genome sequence using Bowtie with zero mismatches, and further calculated the output vi-rus-derived siRNAs (vsiRNAs) with custom perl script and the Linux bash scripts”.
Line 135: What were the evaluation parameters?
Response: Done. In the revised version, we described that “The substitution model was evaluated using ModelTest-NG based on the default pa-rameters [30]”. The substitution model was evaluated using ModelTest-NG based on the default parameters. In the revised version, we have made additional content.
